# Prey–Predator Mathematics Model for Fisheries Insurance Calculations in the Search of Optimal Strategies for Inland Fisheries Management: A Systematic Literature Review

Choirul Basir [1,*] , Asep Kuswandi Supriatna [1,*], Sukono [1] and Jumadil Saputra [2]

1   Departement of Mathematics, Faculty of Mathematics and Natural Sciences, Universitas Padjadjaran, Sumedang 45363, Indonesia; sukono@unpad.ac.id
2   Departement of Economics, Faculty of Business and Social Development, Universiti Malaysia Terengganu, Kuala Nerus 21030, Malaysia; jumadil.saputra@umt.edu.my
*   Correspondence: choirul21001@mail.unpad.ac.id (C.B.); a.k.supriatna@unpad.ac.id (A.K.S.)

**Abstract:** Fish stocking in inland fisheries involves a prey–predator interaction model so that the number of fish stocked affects optimal and sustainable yields. It is very important to make mathematical modeling to optimize inland fisheries management which is part of the blue economy. Currently, studies that focus on predator–prey mathematical modeling in inland fisheries, especially those related to insurance are lacking. The bibliometric database was taken from Google Scholar, Dimensions, Science Direct, and Scopus in the 2012–2022 research years. After further processing, it is displayed on the PRISMA diagram and visualized on VOSviewer to display the update of this research topic. As blue economy sustainability, the management of fisheries sector needs to be reviewed deeply. In this study, the assumptions of the predator–prey mathematical model are made to obtain the equilibrium point, maximum sustainable yield (MSY), and catch per unit effort (CPUE) values. These results can be used to calculate fisheries insurance as a strategy for optimizing sustainable fishermen's income.

**Keywords:** blue economy; fisheries management sector; insurance; prey–predator interaction model; maximum sustainable yield

## 1. Introduction

Inland water areas include river basins and flood plains, lakes, and reservoirs. One of the inland waters that most importantly meet the needs of human life, such as water sources, agriculture, industry, electricity, tourism, and fisheries is a reservoir. Currently, most aquatic ecosystems are experiencing degradation along with the increase in human activities around the reservoir waters, leading to habitat degradation and a decrease in the aquatic environment quality. This has caused changes in the ecological structure of fish resources and a reduction in catches [1,2]. Indirectly, the physical condition of a reservoir is one of the causes of decreased fisheries production. However, this decline is significantly triggered by the high fishing pressure as evidence by the decrease in the catch quota. In order to anticipate the decline in fish production, stocking fish seeds need to be performed as an effort to restore fish resources (fish stock enhancement) [3]. Fish stocking in the reservoir has been carried out previously, and the study results showed that the activity was not optimal because the amount of seed stored was insufficient based on the carrying capacity of the waters [4,5]. Other environmental problems were due to anthropogenic activities around the reservoir waters, predatory fish also tend to prey on smaller ones [6]. This is because each population always interacts with others in an ecosystem, thereby fostering a series of predatory events [7]. According to [8] planktonic and detrivore fish are species suitable for stocking in natural fisheries as they are often preyed upon by predators. Consequently, these prey–predator interactions can affect or alter the state of the fish populations in the ecosystem. It can be concluded that one of the main reasons for the

extinction of prey populations is the very high predation rate or a very low growth rate of the prey [9,10].

Therefore, it is necessary to consider the stocking density settings. Regulating stocking density is generally part of the cultivation work with the media dictating both the size and type of biota it contains. It is important to note that farmers can easily regulate the stocking density in ponds or other fish cultivation media but it is more difficult to estimate the number in natural aquatic ecosystems. This estimation can only be performed through various approaches—one of which is by using a mathematical model. Sufficient studies on stocking densities with mathematical models, which are influenced by commercial fishing aspects, have not been conducted [3,11]. In fact, the model is able to consider other aspects such as: fish stocking, trapping, and interactions between species in a holistic and analytical study. Holistic and analytical models are resourceful in fisheries management. Specifically, the holistic model assists in analyzing historical trends in fisheries while the analytical illustrates the current state of stocks and assesses conditions in subsequent years when fisheries exhibit a similar pattern [12].

Several previous studies underlying this article are the ones on logistic growth models in the field of aquaculture [13,14]. The model described a fishing strategy that can provide optimum harvesting results in order to maintain a sustainable fish population. Another study developed the interaction model of ecologically exploited species [10,14] and the impact of a maximum sustainable catch policy or Maximum Sustainable Yield (MSY) [15]. Furthermore, the prey–predator model of fishing activities in catch-free and reservation areas [16], as well as the dynamics of fisheries resources in the waters have been investigated [16]. Consequently, this current study examined the model of prey–predator populations in reservoirs with additional harvesting efforts for both populations as well as the stocking of prey. The interaction between predators and prey affects the amount of harvest, the prey–predator model obtains an overview of the amount of sustainable harvest or Maximum Sustainable Yield (MSY). The condition of the equilibrium allows the MSY to obtain the optimal number of predators and/or prey to be harvested [15,17]. The MSY value is used as an insurance calculation to provide an overview of the value of the benefits from harvesting so that the insurance calculation is in accordance with sustainable harvesting conditions [18,19].

After obtaining the prey and predator population models, a simulation was conducted on insurance premium calculations for inland fisheries in order to obtain optimal results and mitigate potential significant losses due to unanticipated crop failure or harvests. Successful stocking relies heavily on proper stocking density as deviations from optimal levels can negatively impact physiological processes and behavior, leading to decreased productivity [20]. In addition to their economic value, selected fish species have certain favorable traits such as ease of breeding, rapid growth, resistance to environmental conditions, and can be used as food by the community [16,21]. Based on this phenomenon, a study was conducted to determine the optimal fish stocking level that can enhance production by applying a logistic growth model to analyze fish catches in the reservoir. This current study aims to aid the management of fish resources in reservoir waters through resource restoration. The selection of the reservoir ecosystem was performed to enable effective monitoring and evaluation of the effects of fish stocking [22]. Reservoirs, being classified as closed ecosystems, are often the site of government fish stocking programs.

## 2. Materials and Methods

### 2.1. Scientific Article Data

The current study is built upon those conducted previously, particularly the investigation of logistics growth models in the field of aquaculture [23]. Furthermore, it describes a harvesting approach that provides optimal results while ensuring the preservation and sustainability of fish populations. Meanwhile, another study developed a model that analyzes the interaction between ecologically exploited species and the impacts of MSY policy [24].

Uncontrolled fishing by fishermen can lead to a decline and depletion of fish resources which can be addressed by implementing a proportional fishing strategy [7] and activities periodically or seasonally [23,25]. The exploitation of reservoir fisheries typically occurs through a process of trial and error and while they have been stocked for extended periods. However, the evaluations of stocking effectiveness are infrequently performed [5].

The MSY approach was first introduced by Schaefer in 1954 and 1957. It is a mathematical model that uses a dynamic system-based approach involving the equilibrium point and analyzing both local and global stability. This can ensure the continuity of the system that leads to a successful harvest while maintaining a balanced habitat population [26,27].

### 2.2. Selection of Literature Database

A summary of the search results from the three filtering processes in the four databases can be seen in Table 1.

**Table 1.** The results were positioned in the main text close to their first citation.

| Keywords | Type | Google Scholar | Dimensions | Science Direct | Scopus |
|---|---|---|---|---|---|
| Keywords 1 | A | 1000 | 16,608 | 2324 | 200 |
| Keywords 2 | A AND B | 300 | 292 | 54 | 10 |
| Keywords 3 | A AND B AND C | 12 | 53 | 1 | 0 |

A. ("Predator–Prey Model" OR "Predator-Prey Model" OR "Prey–Predator Model" OR "Prey–Predator Model");
B. ("Fishery") AND ("Maximum Sustainable Yield" OR "MSY");
C. ("Insurance").

Bibliometric search results on Google Scholar used Publish or Perish software for filtering. Meanwhile, for research originating from Dimensions, Science Direct, and Scopus, filtering was carried out on their respective websites with a range of publications from 2012 to 2022. Filtering keywords started from keyword 1 which was typed as A, then it connected "AND" with keywords of type B which produced keywords 2. Finally, from keywords 2 was connected "AND" with keywords of type C which produced keywords 3 as filtering which showed the update of this topic.

After filtering up to keyword 3, the next step was to remove duplicate titles with the help of Jabref software and manually filter abstracts and duplicated text. Table 2 below shows the result of some publications displaying the words under keywords but do not go into detail about the keywords used. This is included in the selection results even though later, the details of the method table to be used are not included because it only appears in words or sentences. However, there is no further discussion of the keywords used as filtering in this process.

**Table 2.** Result selection.

| Database | Data Keywords 3 | Semi-Automatic | | Manual Selection | | | |
|---|---|---|---|---|---|---|---|
| | | Duplicate | | Abstract | | Full Text | |
| | | Excluded | Included | Excluded | Included | Excluded | Included |
| Google Scholar | 12 | 2 | 10 | 4 | 6 | 0 | 6 |
| Dimensions | 53 | 14 | 39 | 15 | 24 | 17 | 7 |
| Science Direct | 1 | 0 | 1 | 1 | 0 | 0 | 0 |
| Scopus | 0 | 0 | 0 | 0 | 0 | 0 | 0 |
| Total | 66 | 16 | 50 | 20 | 30 | 17 | 13 |

The results of the table are displayed in the PRISMA diagram in Figure 1 in order to illustrate the process of filtering articles according to the search.

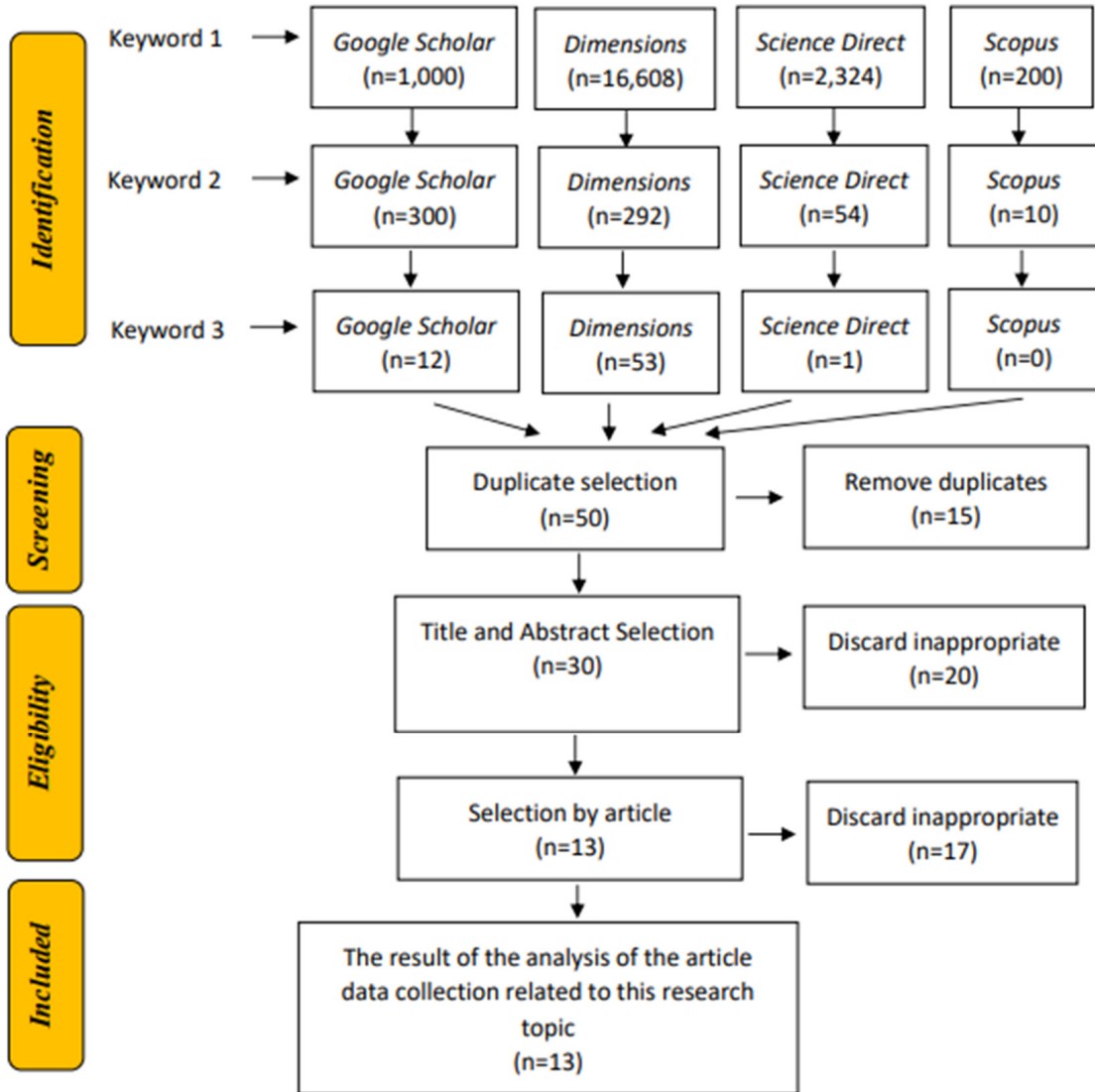

**Figure 1.** PRISMA diagram.

The PRISMA diagram displays the identification process based on three keywords to carry out the screening process for duplication of publications using the Jabref software so that duplication can be removed. At the eligibility stage, filtering was carried out based on duplication of titles and abstracts followed by the selection of the article contents which resulted in 13 publications. The results of the final process include publications that are in accordance with the topic of this research.

### 2.3. Bibliometric Analysis

Articles obtained from the final selection results were then stored in ".RIS" format. Subsequently, the bibliometric analysis, which is a branch of library science for examining information was conducted using VOSviewer software version 1.6.19 was released on January 2023. This technique is commonly used to obtain a scientific bibliographic overview of cited publications through literature. The results of the bibliometric analysis are presented through data visualization, which reveals the relationship between the data in the studied articles. Data visualization aims to analyze the content, patterns, and trends of document collections by measuring term power and counting keywords or topics. The

emerging topics are grouped into several clusters. Each size reflects the number of articles addressing the keywords in the study topic. A large cluster size indicates a widespread discussion of keywords in the database while a small one suggests a limited discussion concerning the theme.

## 3. Results

The following section presents the results of the analysis conducted on 13 articles. This includes a visual representation of article data, namely: the development of prey–predator and Insurance in Fisheries, model analysis, an examination of the methods utilized in model analysis, and numerical simulations.

### 3.1. Article Data Visualization

The topic of prey–predator and insurance in fisheries was visualized using VOSviewer software to determine its novelty (Figure 2).

The visualization demonstrates a limited discourse on the ecological management of fisheries. Meanwhile, the insurance approach within the field remains a noteworthy topic of discussion as it has not become a standard part of the mathematical model.

The discussion of the topic appears to have occurred between 2014 and 2019, although an article search was conducted in the range of 2012–2022 using Publish or Perish. It was observed that the topic of ecology in fisheries has not been explored extensively.

The dimmer visualization is the topic discussion which is rarely carried out while for the bright one there is. However, it can be seen that the density visualization is small. So, it is an interesting topic for further study.

From the density visualization item, the cluster density is displayed to see the topic's relevance. From Figure 2d, it can be seen that the cluster density of the 13 articles is categorized into 4 as shown in the Table 3 below.

From the results of the cluster density visualization topic of prey–predator and insurance in fisheries, a cluster table is shown in Table 3 to show the depth of subject matter for each cluster produced. In general, the topics of each cluster show a good level of density to develop discussions with these topics.

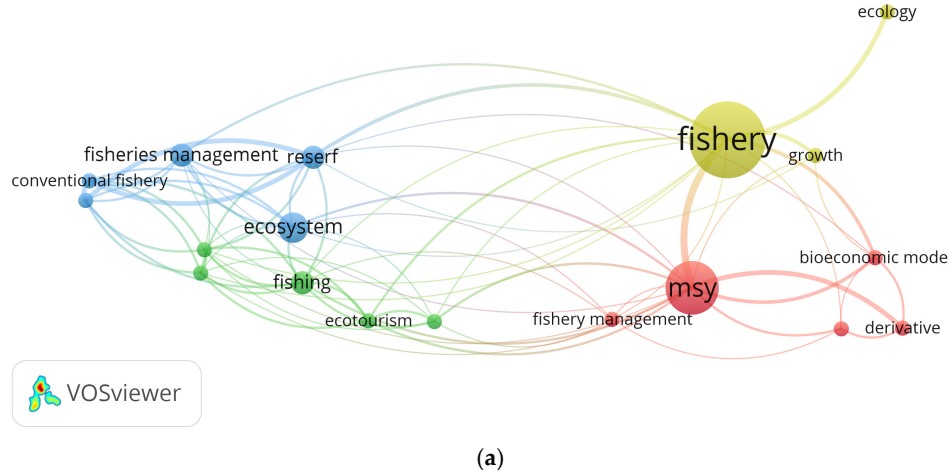

(**a**)

**Figure 2.** *Cont.*

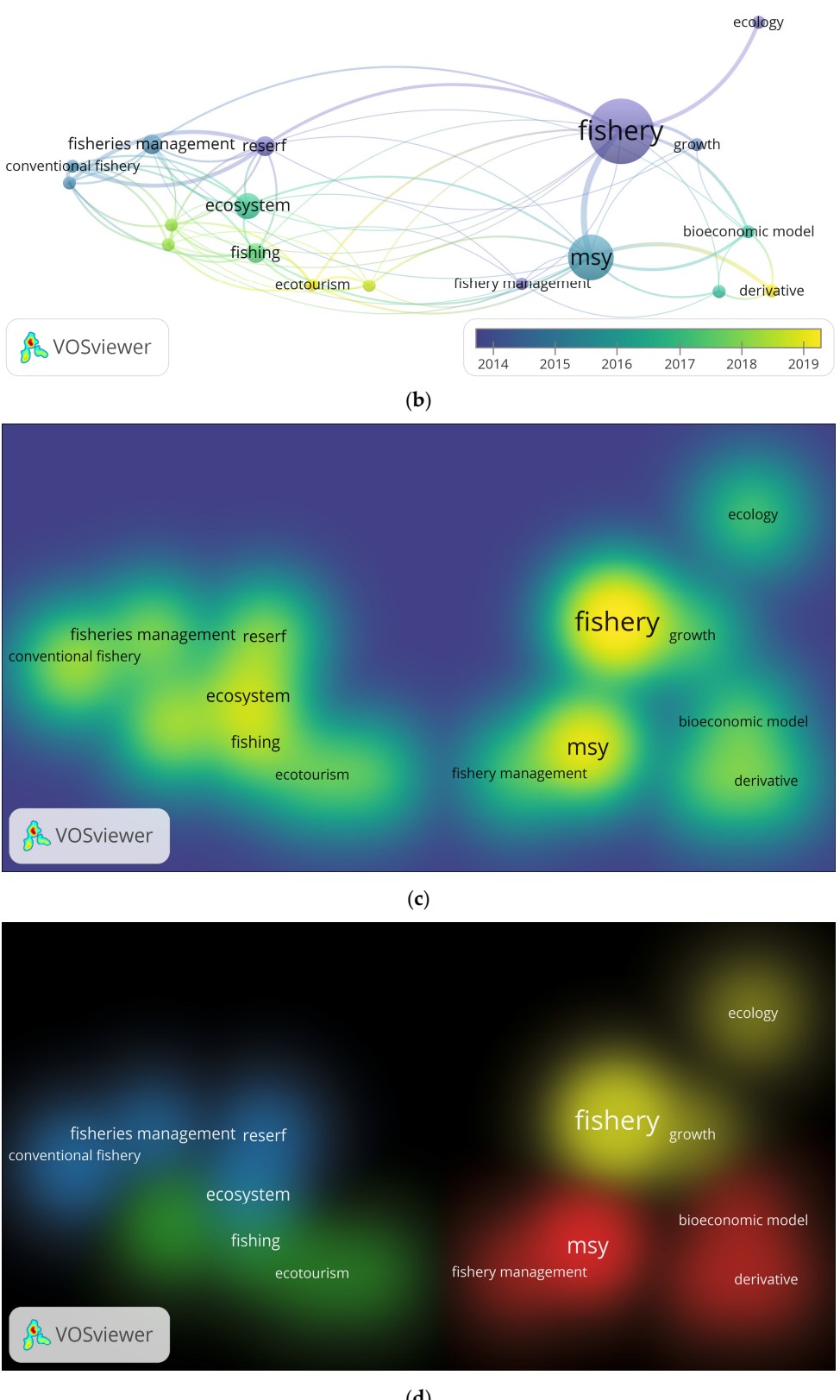

**Figure 2.** (**a**). Network Visualization Topic of Prey–Predator and Insurance in Fisheries; (**b**). Overlay Visualization Topic of Prey–Predator and Insurance in Fisheries; (**c**). Item density Visualization Topic of Prey–Predator and Insurance in Fisheries; (**d**). Cluster Density Visualization Topic of Prey–Predator and Insurance in Fisheries.

**Table 3.** Development of prey–predator and insurance in fisheries.

| Cluster | Items |
|---|---|
| | ecotourism |
| | fishing |
| 1 | impact |
| | predator-prey model |
| | species interaction |
| | conventional fisheries management |
| | ecosystem |
| 2 | fisheries management |
| | insight |
| | reserves |
| | bioeconomic model |
| | derivative |
| 3 | fisheries management |
| | maximum sustainable yield |
| | MSY |
| | ecology |
| | economic |
| 4 | fisheries |
| | growth |

*3.2. Mathematics Model*

From the PRISMA diagram, 13 articles were discussed along with the methods and objects that were the study goals. Not all articles explain the details of the method in detail and thoroughly in the discussion of the article (Table 4).

**Table 4.** Development of prey–predator and insurance in fisheries.

| No | Author | Title | Method | Object |
|---|---|---|---|---|
| 1. | M.D Smith [28] | The new fisheries economics: incentives across many margins | Logistics Models and Lotka–Volterra | Fisheries |
| 2. | Peter Roopnarine [14] | Ecology and the tragedy of the commons | Logistics Model and Lotka–Volterra, Ricker's basic model | General interactions between species |
| 3. | K. Chakraborty, T.K Kar [29] | The economic perspective of marine reserves in fisheries: a bioeconomic model | Logistics Models, and Lotka–Volterra | Fisheries |
| 4. | P. Jakubik [30] | How to anticipate recession via transport indices | Stochastic Dynamics, Lotka–Volterra | Transportation index in fisheries |
| 5. | P. Paul, T.K Kar [31] | Impacts of invasive species on the sustainable use of native exploited species | Logistics Model and Lotka–Volterra | Fisheries |

**Table 4.** *Cont.*

| No | Author | Title | Method | Object |
|----|--------|-------|--------|--------|
| 6. | D. Das, T.K Kar [32] | Marine reserve and its consequences in a predator–prey system for ecotourism and fishing | Single species model, Predator-Prey Model, Harvesting, MSY, EMSY, MEY | Fisheries |
| 7. | X. Chen, G. Li, Q. Ding [25] | A bioeconomic model of fishery resources under ecological and technological interdependencies | Logistics Model and Lotka–Volterra. | Fisheries |
| 8. | A. Gauteplass [33] | On the optimal control of an animal-vegetation ecological system | Logistics Model and Lotka–Volterra, Harvesting, Type II Holling Function Response, MSY | Plants and animals |
| 9. | L.A.K Barnett [34] | Effects of fishing, species interactions, and climate on populations and communities: insights for ecosystem-based fisheries management | Dynamic Models, Fisheries Management, Predator-Prey Model, Harvesting | Fisheries |
| 10. | Seijo, J.C, Defeo, O and Salas, S, FAO [35] | Fisheries bioeconomics. Theory, modeling, and management | Prey–Predator Model | Fisheries |
| 11. | H. Frost, L. Ravensbeck, A. Hoff and P. Andersen. [20] | The economics of ecosystem-based fisheries management | Fisheries Management, MSY, MEY, Ecosystem Dynamic Model, Prey–Predator Model | Fisheries |
| 12. | D. Poudel [36] | Stochastic analysis in fisheries management | Dynamic Growth, Stochastic, Fisheries Management | Fisheries |
| 13. | Tarik C. Gouhier, F. Guichard and Bruce A. Menge. [37] | Designing effective reserve networks for nonequilibrium metacommunities | Logistik Model, Prey–Predator Model, Lotka–Volterra, | Fisheries |

The methods used in the 13 articles from the process shown on the PRISMA flow chart are presented below (Table 5). It was discovered that the studies conducted on the combination of mathematical and insurance models in the context of fisheries are limited in number.

The table reveals that there are no articles in the 2012–2022 publications that discussed insurance in fisheries. Some publications only include keywords but there is no further discussion according to the keywords so that the methods used in the research do not involve all methods in the mathematical model. Most of the 13 articles used the Lotka–Volterra model but only a few used the type II Holling function response method in their research studies. For discussion in material harvesting, not all publications explain in detail, only mentioning the word harvesting without further discussion about harvesting material. Likewise, material discussions about MSY, MEY, and insurance are only shown in sentences but there is no discussion in detail. So, a checklist is not carried out on material discussion.

Visualization of the methods used in the 13 articles resulting from the process shown on the PRISMA flow chart are shown in Figure 3.

**Table 5.** Discussion material of prey–predator and insurance in fisheries.

| No | Author | Title | Discussion Material | | | | | | |
|----|--------|-------|------|------|------|------|-----|-----|-----------|
| | | | Prey–Predator | Lotka–Volterra | Holling Type II | Harvesting | MSY | MEY | Insurance |
| 1. | M.D Smith [28] | The new fisheries economics: incentives across many margins | √ | √ | | | | | |
| 2. | Peter Roopnarine [14] | Ecology and the tragedy of the commons | √ | √ | | | | | |
| 3. | K. Chakraborty, T.K Kar [29] | The economic perspective of marine reserves in fisheries: a bioeconomic model | √ | √ | | | | | |
| 4. | P. Jakubik [30] | How to anticipate recession via transport indices | √ | √ | | | | | |
| 5. | P. Paul, T.K Kar [31] | Impacts of invasive species on the sustainable use of native exploited species | √ | √ | | | | | |
| 6. | D. Das, T.K Kar [32] | Marine reserve and its consequences in a predator-prey system for ecotourism and fishing | √ | √ | | √ | √ | √ | |
| 7. | X. Chen, G. Li, Q. Ding [25] | A bioeconomic model of fishery resources under ecological and technological interdependencies | √ | √ | | | | | |
| 8. | A. Gauteplass [33] | On the optimal control of an animal-vegetation ecological system | √ | | √ | | √ | | |
| 9. | L.A.K Barnett [34] | Effects of fishing, species interactions, and climate on populations and communities: insights for ecosystem-based fisheries management | √ | | | √ | | | |
| 10. | Seijo, J.C, Defeo, O and Salas, S, FAO [35] | Fisheries bioeconomics: Theory, modeling, and management | √ | | | | √ | | |
| 11. | H. Frost, L. Ravensbeck, A. Hoff and P. Andersen. [20] | The economics of ecosystem-based fisheries management | √ | | √ | | √ | √ | |
| 12. | D. Poudel [36] | Stochastic analysis in fisheries management | | | | | | | √ |
| 13. | Tarik C. Gouhier, F. Guichard and Bruce A. Menge. [37] | Designing effective reserve networks for nonequilibrium metacommunities | √ | √ | | | | | |

However, there was also a discussion of the Holling II model, harvesting, and the maximum economic yield (MEY), each accounting for 15% of the discussion. Among the 13 articles reviewed, 92% utilized the prey–predator model and 54% employed the Lotka–Volterra. However, there was discussion of the Holling II, Harvesting, and MEY, each accounting for 15%. Despite being interesting, the discussion on insurance was limited to only a mention of the concept without delving into the specifics of fisheries insurance.

Bar chart visualization of the methods used in the 13 articles resulting from the process shown on the PRISMA diagram are shown in Figure 4.

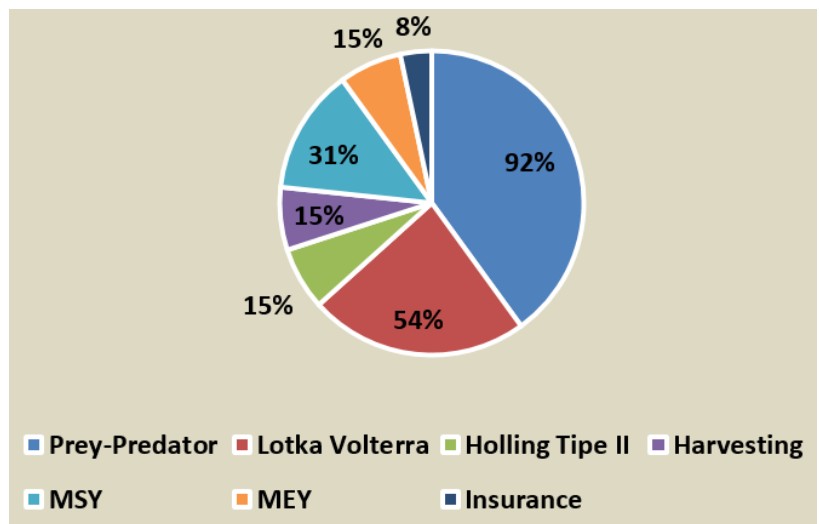

**Figure 3.** Visualization of the methods used in the 13 articles resulting from the process shown on the PRISMA flow chart.

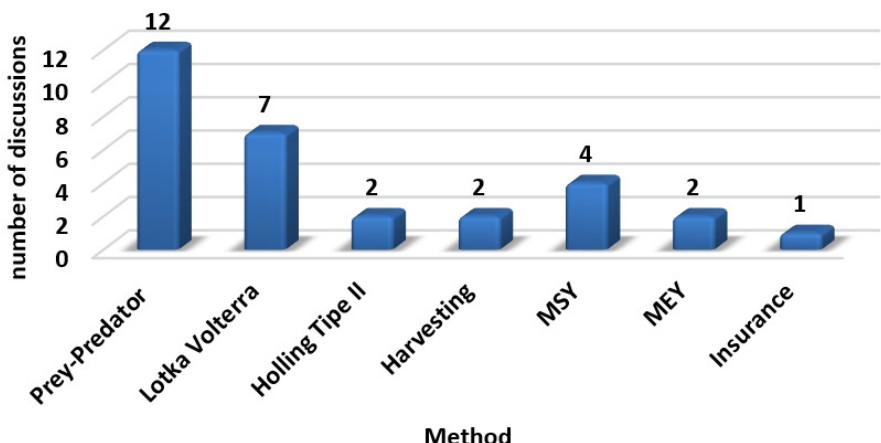

**Figure 4.** Bar chart visualization of the methods used in the 13 articles resulting from the process shown on the PRISMA diagram.

The bar chart shows the article results from the PRISMA diagram discussing in depth the methods of harvesting, MSY, and insurance so that the novelty of research that discusses in depth the methods of prey–predator, harvesting, MSY, and insurance is very good for further research.

In the fisheries sector as a blue economy, insurance can offer a solution for the fishing industry and regulatory entities to tackle some uncertainties and reach their objectives of sustainability, financial security, and increased productivity [18,38]. For example, an insurance company can utilize optimal crop yields to provide insurance services in the field of fisheries, particularly inland fisheries, in order to anticipate and mitigate negative impacts on production results. Since financial institutions can provide a solution in case of adverse circumstances, the inland fisheries industry tends to develop and advance. The government and industry can also enhance the protection and enforcement of regulations in the fisheries sector by establishing and managing funds, utilizing probabilistic forecasts of future catches, prices, and risks from various scenarios, as well as incorporating commercial insurance to preempt the collapse of the fishing industry [18,39].

## 4. Discussion

The use of mathematical models in the context of fisheries is not widespread, particularly when it comes to the topic of fisheries insurance. The development of these models remains broad and varies greatly in the fishing industry, making it a challenging topic to discuss. This study aims to assist the fishing industry in making decisions regarding harvesting and stocking levels. When the models are developed, they can be expanded to include insurance, creating even more diverse mathematical models. This is particularly relevant based on the limited number of articles that have addressed the mathematical modeling of the prey–predator relationship in fisheries, specifically regarding insurance [40,41].

Fishery is a blue economy that must be studied more deeply in its management in order to obtain irrational natural resource conditions. Several mathematical models have actually studied models of interaction between prey and predators [22,42]. However, in terms of application in fisheries as a blue economy, they have not been applied in depth to fisheries management. Therefore, the study of prey–predator mathematical models is one of the solutions that can be applied to fisheries management. For predator–prey mathematical models, several have been studied in research [10,15], but more research should be performed with the existence of a spawning fish population and the presence of predation between spawning fish and predatory fish.

First, consider the traditional prey–predator system which combines selective harvesting efforts on both prey and predator from [43],

$$\begin{aligned} \frac{dx}{dt} &= rx\left(1 - \frac{x}{k}\right) - axy - q_1 e_1 x \\ \frac{dy}{dt} &= bxy - my - q_2 e_2 y \end{aligned} \tag{1}$$

For description of compartments and parameters, see Table 6.

**Table 6.** Description symbol of compartments and parameters.

| Symbol | Description |
| --- | --- |
| $x$ | Prey population |
| $y$ | Predator population |
| $r$ | Constant per capita growth rate |
| $k$ | Constant carrying capacity for the prey species |
| $a$ | The predation rate |
| $b$ | The conversion coefficient due to predation |
| $m$ | The natural mortality rate of the predator species |
| $q_1$ | The catchability coefficients of prey species, respectively |
| $q_2$ | The catchability coefficients of predator species, respectively |
| $e_1$ | The independent harvesting effort on prey species, respectively |
| $e_2$ | The independent harvesting effort on predator species, respectively |

We used the harvesting function based on catch per unit effort (CPUE) hypothesis [10, 44,45].

The mathematical model in the traditional predator–prey system can be developed into a prey–predator mathematical model by involving species that are actually involved in these interactions over time. For example, involving spawning fish from prey fish that also interact with predators. The mathematical model of the interaction between predatory fish with stocking fish and fish spawned from adult fish is a model that will be developed further by involving several parameters. The development of other mathematical models can be carried out without involving predatory fish by implementing a management system that separates prey fish from predatory fish so that they are included in the single

species mathematical model category by harvesting adult fish where they will produce new spawning fish. Stocked fish and spawned fish are physically the same where the difference is only in their origin; stocked fish come from outside the system, while spawned fish come from within the system.

We propose the stages of developing a mathematical model that describes the interaction of four fish populations: stocking fish, fish population in reservoir, spawning fish, and predatory fish. Due to the absence of fishing in the reservoir, the population stocking fish follows a logistic growth model [15]. Fish population in a reservoir increases due to spawning in nature as spawning fish. Fish stocking is carried out as an effort to restore the population because there is continuous fishing, which results in a decrease in fish resources in the reservoir and is assumed to be carried out every year with the value in stocking already determined. This stocking fish population increases the fish density in the reservoir but decreases due to predation of the stocking fish by predatory fish and also due to the natural death. The many studies of prey–predator mathematical modeling on the application of fisheries as a blue economy will be able to provide better and more sustainable fisheries management strategy solutions.

## 5. Conclusions

The implementation of prey–predator mathematical models in addressing problems in inland fisheries is limited, specifically in terms of ensuring sustainable fish harvests. Hence, there is a lack of studies in this area. Also, the discussion of incorporating insurance into these models is limited. There is a need to develop an ideal mathematical model that enables us to meet the maximum sustainable yield (MSY) conditions in order to ensure the sustainable management of inland fisheries as a blue economy. Fishery management as a blue economy that must maintain the sustainability of its harvest must involve mathematics in general, and mathematical models in particular, because in reality there are interactions between species that can be calculated using the prey–predator mathematical model. Mathematical models can be developed from simple models involving single species or multispecies and the parameters are determined based on real conditions in the fishery. For further research, insurance can be included in the variable, not just in an advanced process of a mathematical model. This research does not yet exist, but will include insurance in its mathematical model.

**Author Contributions:** Conceptualization, C.B.; methodology, A.K.S. and S.; formal analysis, J.S. and C.B.; investigation, A.K.S. and S.; resources, A.K.S. and S.; writing—original, C.B. and A.K.S.; writing—revision and editing, C.B.; writing—review and editing, A.K.S. and S.; supervision, C.B. All authors have read and agreed to the published version of the manuscript.

**Funding:** This research is supported by the Indonesian Ministry of Education, Culture, Research, and Technology for Doctoral Dissertation Research Grant 2023, grant number 3018/UN6.3.1/PT.00/2023, entitled "Model Matematika Predator Prey Untuk Perhitungan Asuransi Perikanan Dalam Pencarian Strategi Optimal Pengelolaan Perikanan Darat". The APC was funded by Universitas Padjadjaran.

**Institutional Review Board Statement:** Not applicable.

**Informed Consent Statement:** Not applicable.

**Data Availability Statement:** Not applicable.

**Acknowledgments:** The authors are grateful to the Directorate of Research and Community Service (DRPM) of Universitas Padjadjaran also the Indonesian Ministry of Education, Culture, Research, and Technology.

**Conflicts of Interest:** The authors declared no conflict of interest.

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
