# Peer review of "Prey–Predator Mathematics Model for Fisheries Insurance Calculations in the Search of Optimal Strategies for Inland Fisheries Management: A Systematic Literature Review"

_sustainability, doi:10.3390/su151612376_

Round 1
Reviewer 1 Report
This paper presents an interesting proposal a new survey of Prey Predator Mathematics Model for Fisheries Insurance Calculations in The Search of Optimal Strategies for Inland Fisheries Management. However, there is room for improvement before the manuscript can be accepted for publication. I recommend that the authors carefully revise the manuscript based on the following feedback:
1.The authors should be select exactly of keyword into paper.
2.The introduction should be revised to clearly present the main ideas and motivations behind the proposed research. Please ensure that the research question and motivation of the proposed study are clearly stated. It is important to cover the research gap adequately. And add historical remarks in this section.
3. Add a table of used symbols in the paper to improve readability.
4. The authors should outline of the Survey
5. The authors should Comments on the relevance of the results, comparison of the approaches, critical review of the relevant literature
6. the author should add Open problems in the separation section
Author Response
Dear Reviewer,
I will respond to some revisions from reviewers in this table.
Please see the attachment:
https://drive.google.com/file/d/1p3DXi-X1d4g5Io6dFvDuydQhm3MVgKbU/view?usp=sharing
Thank you for the suggestions that have been given for my paper.
Kind regards,
Choirul Basir

Reviewer 2 Report
Make sure you have a clear and focused research question that guides your search strategy and inclusion criteria.
Make sure to clearly state the research question and objectives of your systematic review in the introduction. This will help guide the reader and provide context for the rest of the review.
Provide a detailed description of the methods used to conduct the systematic review, including the search strategy, inclusion and exclusion criteria, and data extraction and analysis methods. This will help ensure the transparency and reproducibility of your review.
discuss any inconsistencies or discrepancies in the findings and provide possible explanations for them.
Discuss the implications of your findings for fisheries management and insurance calculations. Provide recommendations for future research or policy changes based on your findings.
You can shorten the title to make it more concise and catchy. For example, you can use “Optimal Strategies for Inland Fisheries Management: A Prey-Predator Model with Insurance”.
You can add a section on the background and motivation of your study, explaining why it is important and relevant to address the problem of inland fisheries management and insurance using a prey-predator model.
Use consistent spelling and punctuation throughout the document. For example, use either “predator-prey” or “predator prey” but not both interchangeably.
Use transitions and connectors to link your ideas and paragraphs logically. For example, you could start a new paragraph with “However, …” or “Furthermore, …” to show contrast or addition.
Use active voice instead of passive voice whenever possible. For example, instead of “The results obtained are shown in Table 2.”, you could write “Table 2 shows the results.”
Use appropriate verb tenses and moods to express your meaning accurately. For example, instead of “The fisheries sector as a blue economy should have studied its management more deeply.”, you could write “The fisheries sector as a blue economy needs to study its management more deeply.”
NA
Author Response
Dear Reviewer,
I will respond to some revisions from reviewers in this table.
Please see the attachment:
https://drive.google.com/file/d/1C1zD_bftAhxy8TiXvEytN2WiavVKNvDY/view?usp=sharing
Thank you for the suggestions that have been given for my paper.
Kind regards,
Choirul Basir

Reviewer 3 Report
Reviewer Comments:
Graphical abstract: Not provided or provided with less informative. Suggestion to provide graphical abstract with 3D style so that it is more attractive.
Tables: Try to include more meaningful tables in manuscript.
Figures: Figures are in low resolution. Kindly use higher definition images.
General comment:
This paper is research on mathematical modeling to optimize inland fisheries management. Overall, this paper has well covered a the synthesising of mathematical modeling in inland fisheries. Also, authors need to perform critical analysis and interpret all these studies and come up with a conclusion for each section. It’s good that you had this finding written but readers would preferable want to know what had you concluded from all these studies, instead of what the author of the literature studies had concluded. To conclude, this paper needs to revise it carefully before it can be considered in high impact journal. Hope below comments will be able to help to further improve the paper.
Specific comment:
Abstract:
- Suggest modifying the title to be more attractive and related to the current trends.
- Needs major revision prior to the amendment of the main content.
- An abstract is often presented separately from the article, so it must be able to stand alone. Hence the problem statement, aim, novelty and results of the study, all should be included into the one paragraph of abstract.
- It is suggested that some meaningful result values can be included in this section as well.
- Please try to merge all information into a paragraph with some attractive and new findings.
Keywords:
- Kindly modify the keywords are not the same as the title, because keywords are a tool to help indexers and search engines find relevant papers. If database search engines can find your journal manuscript, readers will be able to find it too. This will increase the number of people reading your manuscript, and likely lead to more citations.
- Please arrange according to alphabetical order.
Introduction:
- Introduction should be covered the gap of the research. However, it is not well covered in this section, it needs to be more specific instead of using only one sentence to cover that.
- Also, please mention the important of this study to society as well as industry.
- Page 1, Line 29-31; kindly rephrase the sentence.
- Page 1, Line 31-34; “…most aquatic ecosystems…”, please do provide some examples on this statement.
- Page 1, Line 36-37; kindly rephrase the sentence.
- Page 1-2, Line 29-50; it is suggested at these two paragraphs to be combine as problem statement for the review.
- Page 2, Line 51; kindly revise “To avoid this scenario above,…” to “To avoid these scenarios…”.
- Page 2, Line 74-84; it is suggested that this paragraph can be moved to the end of this section.
- Page 2-3; Line 85-99; it is suggested that these paragraphs can be either removed or move to other sections.
- Please revised the Introduction section based on the structure below that can make it more clearly:
1st paragraph: Problem statement
2nd paragraph: Current ongoing solution
3rd paragraph: Proposed solution in this work.
4th paragraph: Summarized the current research novelty and objective of this work.
- Objective of your introduction is not strong, need to discuss more about it.
- A good introduction should conclude the introduction by mentioning the specific objectives of the research.
- Kindly refer to those additional materials to revise this section: (1) Evaluating management options for two fisheries that conflict through predator–prey interactions of target species; (2) Statistical design of experimental and bootstrap neural network modelling approach for thermoseparating aqueous two-phase extraction of polyhydroxyalkanoates; (3) Fishery resource management with migratory prey harvesting in two zones- delay and stochastic approach
- The earlier paragraphs should lead logically to specific objectives of the study.
- Note that this part of the Introduction gives specific details: for instance, the earlier part of the Introduction may mention the importance of this study whereas the concluding part will specify what methods of control were used and how they were evaluated.
- At the same time, avoid too much detail because those belong to the Materials and Methods section of the paper.
Materials and Methods:
- Authors need to specify the source of materials were used
- Also, authors are encouraged state the location of the equipment in this report.
- Please elaborate more on the methodology as current description of method is not easy to understand and readers hard to repeat the analysis.
- Statistical analysis is not provided, if possible, try include it in manuscript.
- Please provide an additional figure to illustrate the process of the whole methodology.
Results and discussion sections
- The overall structure needs to be improved
- It is suggested that both results and discussion sections can be combined into single section to ease elaboration/explanation on each results data.
- Very limited information about the detail regarding the listed techniques.
Extensive editing of English language required
Author Response
Dear Reviewer,
I will respond to some revisions from reviewers in this table.
Please see the attachment
https://drive.google.com/file/d/1_DQCfMHg8RD5Er7-mFtkGaUqVlk0w-Xk/view?usp=sharing
Thank you for the suggestions that have been given for my paper.
Kind regards,
Choirul Basir
